# Prevalence and determinants of self-medication consumption of antibiotics in children in Iran: A population-based cross-sectional study, 2018–19

Javad Nazari[1], Nahid Chezani-Sharahi[2], Babak Eshrati[3], Ali Yadegari[4], Mobin Naghshbandi[5], Hamidreza Movahedi[5], Rahmatollah Moradzadeh[6]*

1 Department of Pediatrics, School of Medicine, Arak University of Medical Sciences, Arak, Iran, 2 Arak University of Medical Sciences, Arak, Iran, 3 Department of Community Medicine, School of Medicine, Preventive Medicine and Public Health Research Center, Social Injury Prevention Research Institute, Iran University of Medical Science, Tehran, Iran, 4 Arak Social Tamin Organization, Arak, Iran, 5 Iran University of Medical Sciences, Tehran, Iran, 6 Department of Epidemiology, Faculty of Health, Arak University of Medical Sciences, Arak, Iran

* moradzadehr@yahoo.com, moradzadehr@arakmu.ac.ir

## Abstract

### Introduction

The prevalence of self-medication of antibiotics has been revealed in various studies. The main aim of this work is to investigate the frequency of self-medication in children under 6 years and the factors affecting it.

### Methods

This is a population-based cross-sectional study conducted in the Arak metropolitan in the center of Iran from January 2019 to January 2020. We used stratified random sampling to determine recruitment criteria. As 1754 households were invited to the study that 1483 were approved to participate. Children's data were obtained by the interview with their mothers. In order to define self-medication consumption of antibiotics, it was adapted between annually maternal self-reported consumption of antibiotics among their children and history of received antibiotics registered in insurance services during the same time period. Logistic regression models were exploited to obtain odds ratios and 95% confidence intervals.

### Results

Mean age of mothers was 31.8 years (SD = 5.4), 52.1% of the children were boys. Annually self-medication of antibiotic consumption was estimated 61.6% (n = 914). Based on the logistic regression analysis, in the adjusted analysis, girls were associated with self-medication consumption of antibiotics lower than boys' children (p = 0.016). Older mothers were lower self-medication consumption of antibiotics than youngers (p = 0.001). Moreover, the permanent job of fathers was associated with lower self-medication consumption of antibiotics than temporarily and unemployment (p = .001). The odds of self-medication

**Data Availability Statement:** All relevant data are within the paper and its Supporting Information files.

**Funding:** The author(s) received no specific funding for this work.

**Competing interests:** The authors have declared that no competing interests exist.

consumption of antibiotics were increasing with the increase in age of children (OR: 1.21, CI95%: 1.12, 1.31 and p = 0.001). The increase in parity has been a significant association with the self-medication consumption of antibiotics (OR: 1.64, CI95%: 1.38, 1.95and p = 0.001).

## Conclusion

Results of this study show that some factors such as children's age, gender, mother's age, father's occupational status, and parity are the determinants that significantly impact the self-medication prevalence.

## Introduction

Self-medication is defined as the utilization of drugs to treat self-diagnosed disorders or symptoms, and also irregular or continuous use of a prescribed drug for chronic or repeated symptoms of diseases [1]. According to the World Health Organization(WHO), in order to achieve rational drug use, it is necessary to use appropriate drugs that can eliminate the clinical needs of patients in a specific geographical area with the least complications and costs [2, 3].

On average, antibiotics make up from 30 to 50% of prescription drugs. Although antibiotics are necessary in most bacterial infections and non-prescription can be life-threatening in some cases, most studies have shown that prescribing antibiotics in 30% of prescriptions is incorrect and in 60% are inappropriate, and these errors are usually done by either the physicians or patient's self-medication [4, 5]. There is always the concern that new strains of microbes won't be treated with existing antibiotics, causing mortality in patients infected with these new strains. The highest incidence of resistance to antibiotics has been observed in the countries with the highest consumption of antibiotics, thus, new and more effective policies need to be implemented to control the overuse of antibiotics [6–8].

In 1990, antibiotics accounted for 12% of all drugs sales worldwide; 19% percent in developing countries, which reached 34% in 2000. Economically, the cost of antibiotic treatment in 2000 was estimated to be 40 billion dollars, one-third of which was in developing countries. Thus, despite efforts taking place, the growth in the antibiotic's price, especially in developing countries, has continued [9]. A study by Eili Y. Klein et al. which looked at the global increase and geographic convergence of antibiotics usage between 2000 and 2015 in 76 countries, found that antibiotics usage increased by about 39 percent between mentioned years. This increase was more prominent among middle to lower-income countries [10].

The prevalence of self-medication of antibiotics has been distinct in different countries, these figures are as follows: Sudan (48.1%), Spain (41%), Greece (74.6%), Malta (30%) and Iran (83%), As can be seen, this statistic is higher in Iran than in many other developing countries. This is a matter of concern and it seems that more studies are needed on this issue and the factors affecting it at the community level [8, 11–15].

Despite various studies on the usage of antibiotics in different countries at various levels of society, only sporadic studies in this regard have been conducted in Iran, most of them investigated antibiotic resistance and the types of microbial resistant strains, and there are very few population-based studies, that investigate how to use antibiotics. Therefore, the prevalence of self-medication of antibiotics in Iran and the affecting factors was surveyed using the above-mentioned method. In accordance with this report and the registered insurance sources, it was

determined whether the consumption was self-medication or was done according to the physician's prescription.

Lack of coordination in prescribing drugs by physicians in our country is a serious issue and implementing careful policies in the field of prescribing antibiotics seems necessary, but before that, full knowledge of the details of the pattern of prescription and consumption of antibiotics is important [16].

Due to the underdevelopment of their immune system, children are susceptible to many diseases and infections, especially viral and bacterial types. More than 80% of these infections are viral and have many similarities to bacterial infections in clinical signs and symptoms. Self-medication is a critical health concern that might cause several problems for children such as antibiotic-induced drug resistance, raised drug use per capita, non-desired treatment, and drug toxicity. Studies have shown that self-medication of antibiotics is accounted for 3% of congenital anomalies. In many patients, it has been observed that the use of antibiotics is decided by the family or given to them by the pharmacy. We must train people of the community not to take antibiotics without a physician's prescription and inform them about the side effects of overuse of antibiotics.

Regarding the prevalence of self-medication of antibiotics in Iran, it seems that such studies like this are necessary and it is important to determine what steps can be taken to improve the rational use of antibiotics. Therefore, the purpose of this study is to investigate the frequency of self-medication in children under 6 years and factors affecting it, as to be the basis of controlling to use of antibiotics and subsequent complications in the city of Arak.

## Methods

This is a population-based cross-sectional study conducted in the Arak metropolitan in the center of Iran from January 2019 to January 2020. We used stratified random sampling to determine the recruitment criteria. There are 50 centers to provide health care in the whole of Arak that cover the whole population of the city. These centers were organized study strata. Among strata, the study samples were selected proportion to size by simple random sampling from the covered population. Finally, 1754 households were invited to the study that 1483 were approved to participate. The response rate obtained was 85%.

Inclusion criteria were having Iranian citizenship, having lived in Arak for at least 3 years, having at least one child aged between0-5 years old. The provided questionnaires were completed by interviews. Children's data were obtained by the interview with their mothers.

In this study, we asked families whether or not they had used antibiotics for their children in the past year, either with physicians prescription or self-medicated. In order to define self-medication consumption of antibiotic, it was adapted between annually maternal self-reported consumption of antibiotic among their children and history of received antibiotics registered in insurance services in same time duration, so any maternal self-reported consumption of antibiotic that was not reported in insurance services was considered as self-medication of antibiotics.

Other variables that included in this study were the age of children and mother (years) and gender (male/female) of children, mother educational level (illiterate and preliminary, guidance school, high school, diploma, higher than a diploma, undergraduate and postgraduate), father educational level (illiterate and preliminary, guidance school, high school, diploma, higher than a diploma, undergraduate and postgraduate), mother occupational status (unemployment, housekeeper, temporary job, permanent job), father occupational status (unemployment, temporary job, permanent job), parity (count), insurance coverage (yes/no) and socio-economic status (SES) (1st to 5th quintiles based on assets). SES was obtained based on the assets including owning a usual refrigerator, side by side refrigerator, color TV, LCD TV, LED

TV, cell phone, washing machine, dishwasher, microwave, vacuum cleaner, motorcycle, laptop, access to the internet at home, car, home, number of rooms for sleeping, sanitation facility, bathroom, cooling equipment, heating devices, cooking equipment and place of cooking at home [17]. Health literacy was assessed by Health Literacy for Iranian Adults (HELIA) questionnaire [18] on a 5-point Likert scale. To calculate the total score for HELIA, a scoring manual based on a study by Tavousi et al. was used [18].

### Ethics

Ethic approval was achieved from the Ethics Committee of Arak University of Medical Sciences with the code of ethics as following: IR.ARAKMU.REC.98.217. A written informed consent was obtained from all participants. Participants were informed that they are free to participate and can withdraw from the study at any time.

### Statistical analysis

Frequency, mean and standard deviation (SD) were calculated for the categorical and continuous variables. SES based on asset indices was obtained by principal component analysis [17]. Chi-square and Mann-Whitney U test were used to compare mean or median values by self-medication of antibiotic consumption. Univariate and multiple logistic regression models were applied to obtain odds ratio (OR) and 95% confidence intervals. In the final model of the multiple logistic regression model, the variables with p-value > 0.2 were excluded. All analysis was applied in SPSS version 16.0 and STATA version 12.0 software. P-values < 0.05 were considered to indicate statistical significance.

### Results

Descriptive characteristics of the participants were shown in Table 1. The mean age of mothers was 31.8 years (SD = 5.4). 52.1% of the children were boys. Most of the mothers' and fathers' educational levels were diplomas (38% and 37.8%, respectively). Occupation of the majority of the mothers and fathers were in order of housekeeper (88.2%) and permanent works (61.0%). Most mothers had a maximum of two children (89.2%). The mean of birth weight of children was 3174 grams (SD = 466.7). Insurance coverage was also found among 93.1% of the participants.

Annually self-medication of antibiotic consumption was estimated 61.6% (n = 914). Furthermore, in chi-square and Mann-Whitney analysis, there were statistically significant differences in self-medication of antibiotic consumption by children sex (p = 0.013), mother education (p = 0.042), father job (p = 0.013) and parity (p = 0.001).

Based on the logistic regression analysis, univariate and adjusted findings were shown in Table 2. Finally, in the adjusted analysis, girls were associated with lower self-medication consumption of antibiotics than boys (p = 0.016). Older mothers were lower self-medication consumption of antibiotics than youngers (p = 0.001). Moreover, the permanent job of fathers was associated with lower self-medication consumption of antibiotics than temporarily and unemployment (p = .001). The odds of self-medication consumption of antibiotics were increasing with an increase in the age of children (OR: 1.21, CI95%: 1.12, 1.31 and p = 0.001). The increase in parity has been a significant association with the self-medication consumption of antibiotics (OR: 1.64, CI95%: 1.38, 1.95and p = 0.001).

### Discussion

Nowadays, the overuse of antibiotics is one of the most important health issues in various countries from developed to developing countries. Their side effects, like any other drug, can

**Table 1. Demographic characteristics of the maternal self-medication consumption of antibiotic among children under 6 years, 2018–2019.**

| Variables | | N (%) | Self-medication consumption of antibiotic | | |
|---|---|---|---|---|---|
| | | | No (%) | Yes (%) | P-value |
| Mother age | Mean (SD) | 31.8 (5.4) | 32.1(5.3) | 31.6(5.5) | 0.094[b] |
| Children sex | boy | 772 (52.1) | 276 (48.5) | 496 (54.3) | 0.031[a] |
| | girl | 711 (47.9) | 293 (51.5) | 418 (45.7) | |
| Mother education | Illiterate and preliminary | 99(6.7) | 39(6.9) | 60(6.6) | 0.042[a] |
| | Guidance school | 183(12.3) | 59(10.4) | 124(13.6) | |
| | High school | 60(4.0) | 15(2.6) | 45(4.9) | |
| | Diploma | 564(38.0) | 209(36.7) | 355(38.8) | |
| | Higher diploma | 115(7.8) | 50(8.8) | 65(7.1) | |
| | undergraduate | 396(26.7) | 171(30.1) | 225(24.6) | |
| | postgraduate | 66(4.5) | 26(4.6) | 40(4.4) | |
| Father education | Illiterate and preliminary | 102(6.9) | 33(5.8) | 69(7.5) | 0.141[a] |
| | Guidance school | 188(12.7) | 60(10.5) | 128(14.0) | |
| | High school | 66(4.5) | 30(5.3) | 36(3.9) | |
| | Diploma | 560(37.8) | 210(36.9) | 350(38.3) | |
| | Higher diploma | 149(10.0) | 58(10.2) | 91(10.0) | |
| | undergraduate | 302(20.4) | 128(22.5) | 174(19.0) | |
| | postgraduate | 116(7.8) | 50(8.8) | 66(7.2) | |
| Mother job | unemployment | 6(0.4) | 3(0.5) | 3(0.3) | 0.598[a] |
| | housekeeper | 1308(88.2) | 498(87.5) | 810(88.6) | |
| | Temporary job | 45(3.0) | 15(2.6) | 30(3.3) | |
| | Permanent job | 124(8.4) | 53(9.3) | 71(7.8) | |
| Father job | unemployment | 15(1.0) | 3(0.5) | 12(1.3) | 0.013[a] |
| | Temporary job | 563(38.0) | 194(34.1) | 369(40.4) | |
| | Permanent job | 905(61.0) | 372(65.4) | 533(58.3) | |
| Parity | 1 | 692(46.7) | 1.5(0.7) | 1.7(0.7) | 0.001[b] |
| | 2 | 631(42.5) | | | |
| | 3 | 138(9.3) | | | |
| | 4 | 19(1.3) | | | |
| | 5 | 3(0.2) | | | |
| Birth weight | Mean (SD) | 3174(466.7) | 3183(455.1) | 3168(473) | 0.755[b] |
| Insurance coverage | Yes | 1381(93.1) | 526(92.4) | 855(93.5) | 0.415[a] |
| | No | 102 (6.9) | 43(7.6) | 59(6.5) | |
| Socio-economic status | 1st quintile | 297(20) | 118(20.7) | 179(19.6) | 0.077[a] |
| | 2nd quintile | 297(20) | 113(19.9) | 184(20.1) | |
| | 3rd quintile | 296(20) | 102(17.9) | 194(21.2) | |
| | 4th quintile | 297(20) | 104(18.3) | 193(21.1) | |
| | 5th quintile | 296(20) | 132(23.2) | 164(17.9) | |
| Health literacy | Mean (SD) | 70.16(17.55) | 70.15(17.66) | 70.17(17.38) | 0.775 |

affect people's health in several ways. In addition, their excessive consumption can lead to microbial resistance in community members. Moreover, interactions of antibiotics with other drugs can cause irreversible side effects for antibiotic consumers without real necessity. Self-medication of antibiotics, especially if accompanied by the incorrect dosage, can thwart treatment of curable infections in the future.

Today, the role of the individual as a central factor in health management is emphasized. Terms such as patient centralization, lifestyle, patient actions, and empowerment all emphasize

**Table 2. Association between maternal self-medication of antibiotics for children and independent variables in univariate and adjusted logistic regression analysis among children 0–5 years old, 2018–2019.**

| Variable | Univariate analysis | | Adjusted analysis* | |
|---|---|---|---|---|
| | OR (95%CI)** | P-value | OR (95%CI)** | P-value |
| Children sex (girl) | 0.79(0.64, 0.98) | 0.031 | 0.77 (0.62, 0.95) | 0.016 |
| Age of children | 1.18(1.09, 1.27) | 0.001 | 1.21(1.12, 1.31) | 0.001 |
| Age of mother | 0.99(0.97, 1.01) | 0.207 | 0.96(0.94, 0.98) | 0.001 |
| Mother education | 0.92(0.86, 0.98) | 0.015 | - | - |
| Father education | 0.92(0.86, 0.98) | 0.010 | - | - |
| Mother job | 0.94(0.78, 1.12) | 0.465 | - | - |
| Father job | 0.74(0.60, 0.91) | 0.004 | 0.80(0.65, 0.99) | 0.046 |
| Parity | 1.48(1.26, 1.72) | 0.001 | 1.64(1.38, 1.95) | 0.001 |
| Birth weight | 1.00(0.99, 1.01) | 0.545 | - | - |
| Insurance coverage | 1.18(0.79, 1.78) | 0.415 | - | |
| Socio-economic status | 0.97(0.90, 1.05) | 0.455 | - | - |
| Health literacy | 1.00(0.99, 1.01) | 0.985 | - | - |

that the individuals have a vital role more than the health care provider in controlling their health and they can participate in its healthcare decisions [19]. Some factors such as the production of new drugs, the growth of various diseases, access to health information through the Internet, are important factors that in recent decades have increased the possibility for people to participate in health decisions and management themselves. this issue has led to an increase in the importance of health literacy in an individual's life [20]. Health literacy should be considered as a set of cognitive and social skills that can determine the motivation and ability of individuals to access, insight, and use the information that a person achieves and uses in a way that one can maintain and promote one's health [21]. Although it is not yet clear exactly how much health literacy affects human health outcomes, there are many reasons why many of the unfortunate health outcomes are the result of inadequate health literacy in different societies.

It is observed in different countries, both developed and underdeveloped, the level of health literacy can affect the level of public health, for example in a country like the United States, a meta-analysis study which summarized the results of 85 different studies, showed that the percentage of inadequate and borderline health literacy in the United States was estimated between 20 and 25 percent [22]. The study by Wizadi et al. in 1394, showed that people with better health literacy, could assess their general health status more suitably, they also took more prevention behaviors than others [23].

In a study conducted in Tehran in 1998 on prescriptions and their amount of antibiotics, it was found that in 43% of prescriptions, antibiotics were prescribed [24]. Also, in another study of the pattern of antibiotic use in Taleghani Hospital in Tehran, it was observed that about 57% of the patients had received antibiotics [25]. The first few years of life, despite the small share it has in the whole human life, will have a large contribution to health in adulthood. Antibiotic treatment can cause many problems in children, for instance, physicians should consider the types of common infectious diseases at different ages, their specific antibiotics, and the specific dosage therapy of each agent, and the toxicity of the drugs. This indicates that the use of antibiotics needs more serious attention in future studies.

A cross-sectional study that was conducted in 2013 by Francesco et al. showed that only 10% of people had a correct definition of antimicrobial resistance, 20% of them were familiar with the proper use of antibiotics, and one-third of them had self-medicated antibiotics last year. Finally, the researchers of this study suggested holding training courses to increase the level of awareness and improve community performance concerning antibiotic use [19].

Another study by Belongia et al. in 1999, also suggests providing information and educational messages about antibiotics for people, resulting in an enhancement in the knowledge of people and reduction in patients' demand for antibiotics [26]. In a study conducted by Greek researchers, a very low percentage of patients(0.4%) had taken antibiotics as self-medication (22), Although this study about the use of antibiotics in the community during the last year was done through a questionnaire, but the target population was selected from the parents of children hospitalized, which can be biased in the selection of the study population and does not reflect the situation of the whole community, Perhaps the difference in the results regarding the prevalence of antibiotic use in this study with our survey is due to these differences that existed in the selection of demographic samples in the two studies.

In accordance to our findings, which resulted in that the rate of antibiotic self-medication was increased by the mothers age, the study by Sayer I. Al-Azzam et al. on the overuse of antibiotics and self-medication in Jordan showed that the rate of self-medication was significantly affected by age ($P < 0.001$), income ($P = 0.037$), and educational level ($P < 0.001$), but not by gender ($P = 0.528$) [27]. In this survey, which has been planned at the community level, the self-medication and the factors affecting have been studied, and among the effective factors, the level of education, age, and income have been associated significantly. However, in a study by Vaananen & colleagues on Finnish immigrants in southern Spain in 2002, The results indicated that there are no statistical differences between gender, age, marital status, working situation, self-reported health, or smoking of antibiotic consumers, which was inconsistent with our obtained results [8]. In this study, different statistics have been obtained from our research on self-medication, which in itself can confirm that the issue of self-medication of antibiotics in different communities and countries gives distinct results. Whilst in this study, unlike other studies, no significant relationship was observed between various individual and social factors and the rate of self-medication antibiotics. We believe that the difference seen among results various studies may be occurred due to the effect of different geographical, cultural and historical factors affecting each region.

In a cross-sectional study by Zolali et al., Antibiotic prescription was correlated to the current diagnosis, mother education, and parents perception score for the proper antibiotic. There were certain factors defined as protective and risky affecting antibiotic prescription. Protective factors included physician specialty mother education and work status. Risk factors included parents' perception of antibiotics and the presence or absence of fever [28]. In this survey as in our study, sampling was performed based on the community level using valid questionnaires on the population of children, with the difference that the samples selected for the survey were among outpatients of the hospital clinic, which may cause bias in this survey, while in our study, the samples were randomly selected from the community and the questionnaires were completed by trained individuals referring to their home.

## Conclusion

In conclusion, our findings indicate a remarkable prevalence of antibiotic self-medication among children aged between 0–6 in Iran. Elements recognized as determinants that could potentially impact the self-medication prevalence consisted of children's age, gender, mother's age, father's occupational status, and parity. Further studies are suggested to designate other effective agents on self-medication prevalence and methods to reduce it.

## Suggestions

These results clearly show that to correct the irregular use of antibiotics, basic measures should be taken at all levels, from society level to pharmacy staff and physicians and pharmacist. Also,

measures should be planned in order to raise the level of health literacy of the people in this regard. In addition, it seems that the establishment of serious deterrent laws and efforts to implement them accurately and completely can pave the way for the improvement of the existing process, which has led to the irrational and over use of medications, especially antibiotics in our community. The health system should be organized to improve the universal access, offering the essential care and the essential medicines required according to the population needs. It is needed to rethink how the health interventions should be planned and performed, especially at the primary care setting.

## Supporting information

**S1 Data. Minimal child antibiotic.**
(SAV)

## Author Contributions

**Conceptualization:** Ali Yadegari.

**Investigation:** Ali Yadegari.

**Methodology:** Nahid Chezani-Sharahi.

**Project administration:** Javad Nazari, Rahmatollah Moradzadeh.

**Supervision:** Javad Nazari, Babak Eshrati, Rahmatollah Moradzadeh.

**Validation:** Babak Eshrati.

**Writing – original draft:** Mobin Naghshbandi, Hamidreza Movahedi.

**Writing – review & editing:** Mobin Naghshbandi, Hamidreza Movahedi.

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
