## [Decision Letter · Decision Letter 0]

26 Jan 2022

PONE-D-21-31099Prevalence and determinants of self-medication consumption of antibiotics in children in Iran: a population-based cross-sectional study, 2018-19PLOS ONE

Dear Dr. Moradzadeh,

Thank you for submitting your manuscript to PLOS ONE. After careful consideration, we feel that it has merit but does not fully meet PLOS ONE’s publication criteria as it currently stands. Therefore, we invite you to submit a revised version of the manuscript that addresses the points raised during the review process.

We look forward to receiving your revised manuscript.

Kind regards,

Hans-Uwe Dahms, Ph.D.

Academic Editor

PLOS ONE

Journal Requirements:

a) Did participants provide their written or verbal informed consent to participate in this study?

5. Please ensure that you include a title page within your main document. You should list all authors and all affiliations as per our author instructions and clearly indicate the corresponding author.

Additional Editor Comments:

This contribution needs major changes as indicated by the 2 reviewers.

Reviewers' comments:

Reviewer's Responses to Questions

**Comments to the Author**

1. Is the manuscript technically sound, and do the data support the conclusions?

Reviewer #1: Yes

Reviewer #2: Yes

2. Has the statistical analysis been performed appropriately and rigorously? 

Reviewer #1: Yes

Reviewer #2: Yes

3. Have the authors made all data underlying the findings in their manuscript fully available?

Reviewer #1: Yes

Reviewer #2: Yes

4. Is the manuscript presented in an intelligible fashion and written in standard English?

Reviewer #1: Yes

Reviewer #2: Yes

5. Review Comments to the Author

Reviewer #1: Paper Review

Prevalence and determinants of self-medication consumption of antibiotics in children in Iran: a population-based cross-sectional study, 2018-19

The main objective of this study is to investigate the frequency of self medication in children under 6 years and the factors affecting it. Self-medication is utilization of drugs to treat self-diagnosed disorders or symptoms, and also irregular or continuous use of a prescribed drug for chronic or repeated symptoms of diseases. This study uses Chi- 5 square and Mann-Whitney U test were used to compare mean or median values by self-medication of antibiotic consumption. Univariate and multiple logistic regression models were used and p < 0.05 were considered. This study was in Arak metropolitan in the center of Iran from January 2019 to January 2020 and 1754 households were invited to the study that 1483 were approved to participate.

This paper analyses factors such as children’s age, gender, mother’s age, father’s occupational status, and parity .

Key Results Obtained:-

Annually self-medication of antibiotic consumption = 61.6% (n = 914).

Mean age of mothers = 31.8 years (SD = 5.4)

Older mothers were lower self-medication consumption of antibiotics than youngers (p = 0.001)

Association of girls with self-medication consumption of antibiotics was lower than boys (p = 0.016)

Chi-square and Mann-Whitney analysis, gave the results significant differences in selfmedication of antibiotic consumption by children sex (p = 0.013), mother education (p = 0.042), father job (p = 0.013) and parity (p = 0.001).

Permanent job of fathers was associated with lower self-medication consumption of antibiotics than temporarily and unemployment (p = .001)

Odds of self-medication consumption of antibiotics were increasing with an increase in the age of children (OR: 1.21, CI95%: 1.12, 1.31 and p = 0.001)

Increase in parity shown a significant association with the self-medication consumption of antibiotics (OR: 1.64, CI95%: 1.38, 1.95and p = 0.001).

Positive feedback

The factors considered for the study are effective , valid and as the samples were selected randomly , study was unbiased.

The models and softwares which used were updated and the study considered most of the points which are important in investigating the factors affecting self – medication consumption of antibiotics.

Critical Comments

Others factors such as production of new drugs, the growth of various diseases, access to health information through the Internet could have been considered diring study cause they play important role in health literacy.

Measures to increase the level of awareness and improve community performance about antibiotic use could be suggested

Reviewer #2: The article needs fundamental changes. please pay attention:

This study is about self-medication in children, but in the introduction, not enough explanation is given about the target group, for example, explain what are the side effects of self-medication in children? Explain the negative consequences of self-medication in children.

Was the questionnaire valid and reliable? Give a full explanation.

The final conclusion is a very general written and repetitive text of the results.

Add restrictions and suggestions to the article.

At the end of the article, suggest solutions to reduce self-medication in children.

6. PLOS authors have the option to publish the peer review history of their article (what does this mean?). If published, this will include your full peer review and any attached files.

Reviewer #1: No

Reviewer #2: No

---

## [Author Response · Author response to Decision Letter 0]

25 Mar 2022

Thank you for the comments on our manuscript entitled " Prevalence and determinants of self-medication consumption of antibiotics in children in Iran: a population-based cross-sectional study, 2018-19". We appreciate the suggested modifications and have revised the manuscript accordingly. The revised sections are shown in boldface type and written with different color. The detailed responses to the reviewers’ comments are presented as follows:

Journal Academic Editor

Comment 1: Please ensure that your manuscript meets PLOS ONE's style requirements, including those for file naming

Response: Thanks for your concern upon style requirements, the mentioned naming style requirements of the files will be fulfilled upon re-submission process.

Comment 2: Please amend your current ethics statement to address the following concerns:

a) Did participants provide their written or verbal informed consent to participate in this study?

Response: A written informed consent was obtained from the participants of the study. The full ethics statement including the ethic codes and criteria of the informed consent will be available in the renewed manuscript in the paragraph entitled “Ethnicity criteria”, located in the first paragraph of the page No 5.

Comment 3: In your Data Availability statement, you have not specified where the minimal data set underlying the results described in your manuscript can be found. PLOS defines a study's minimal data set as the underlying data used to reach the conclusions drawn in the manuscript and any additional data required to replicate the reported study findings in their entirety. All PLOS journals require that the minimal data set be made fully available.

Response: In our re-submission, this issued will be addressed via providing a supporting additional file consisting of the minimal data set of the results of our study. Thanks for your concern.

Comment 4: PLOS requires an ORCID iD for the corresponding author in Editorial Manager on papers submitted after December 6th, 2016. Please ensure that you have an ORCID iD and that it is validated in Editorial Manager.

Response: The ORCID iD of the corresponding author will be available in the re-submission process and also on the renewed title page.

Comment 5: Please ensure that you include a title page within your main document. You should list all authors and all affiliations as per our author instructions and clearly indicate the corresponding author.

Response: Thanks, a renewed title paged including the authors names and affiliations will be uploaded upon the re-submission.

Comment 6: Please include your full ethics statement in the ‘Methods’ section of your manuscript file. In your statement, please include the full name of the IRB or ethics committee who approved or waived your study, as well as whether or not you obtained informed written or verbal consent. If consent was waived for your study, please include this information in your statement as well. 

Response: The full ethics statement including the ethic codes and criteria of the informed consent will be available in the renewed manuscript in the paragraph entitled “Ethnicity criteria”, located in the first paragraph of the page No 5.

Reviewer 1:

Comments: Positive feedback

The factors considered for the study are effective , valid and as the samples were selected randomly , study was unbiased.

The models and softwares which used were updated and the study considered most of the points which are important in investigating the factors affecting self – medication consumption of antibiotics.

Critical Comments

Others factors such as production of new drugs, the growth of various diseases, access to health information through the Internet could have been considered diring study cause they play important role in health literacy.

Measures to increase the level of awareness and improve community performance about antibiotic use could be suggested.

Response: Thanks for your elegit and precise comments, the necessary changes are applied to the reviewed manuscript. Proposed corrective measures are mentioned in the “Suggestions” section located as the last paragraph of the page No 10.

Reviewer 2:

Comments: This study is about self-medication in children, but in the introduction, not enough explanation is given about the target group, for example, explain what are the side effects of self-medication in children? Explain the negative consequences of self-medication in children.

Was the questionnaire valid and reliable? Give a full explanation.

The final conclusion is a very general written and repetitive text of the results.

Add restrictions and suggestions to the article.

At the end of the article, suggest solutions to reduce self-medication in children.

Response: Thanks for your concern,

 A more detailed explanation of the side effects of self-medication of antibiotics in children is added to the Introduction section.

By referring to the questionnaire, we believe that you are referring to HELIA (Health Literacy for Iranian Adults) questionnaire. Numerous previously published studies suggest that the Health Literacy for Iranian Adults (HELIA) is a reliable and valid instrument for measuring health literacy in Iran.(1-4)

The conclusion section has been changed, thanks for your apprehension.

Suggestions for solutions to reduce self-medication has been added to the renewed manuscript in the paragraph entitled “Suggestions” located

References

1. Chahardah-Cherik S, Gheibizadeh M, Jahani S, Cheraghian B. The relationship between health literacy and health promoting behaviors in patients with type 2 diabetes. International journal of community based nursing and midwifery. 2018;6(1):65.

2. Khoshnudi M, Safari A, Vahedian-Shahroodi M, Sadeghnejhad H, Nejati Parvaz N. The Relationship between Health Literacy and Quality of Life in Nurses of hospitals of Kashmar in 2018. Journal of Health Literacy. 2019;4(1):9-17.

3. Montazeri A, Tavousi M, Rakhshani F, Azin SA, Jahangiri K, Ebadi M, et al. Health Literacy for Iranian Adults (HELIA): development and psychometric properties. 2014.

4. Tavousi M, Haeri-Mehrizi A, Rakhshani F, Rafiefar S, Soleymanian A, Sarbandi F, et al. Development and validation of a short and easy-to-use instrument for measuring health literacy: the Health Literacy Instrument for Adults (HELIA). BMC public health. 2020;20(1):1-11.

---

## [Decision Letter · Decision Letter 1]

25 Oct 2022

PONE-D-21-31099R1Prevalence and determinants of self-medication consumption of antibiotics in children in Iran: a population-based cross-sectional study, 2018-19PLOS ONE

Dear Dr. Moradzadeh,

Thank you for submitting your manuscript to PLOS ONE. After careful consideration, we feel that it has merit but does not fully meet PLOS ONE’s publication criteria as it currently stands. Therefore, we invite you to submit a revised version of the manuscript that addresses the points raised during the review process.

We look forward to receiving your revised manuscript.

Kind regards,

Ugurcan Sayili, M.D.

Academic Editor

PLOS ONE

Journal Requirements:

Reviewers' comments:

Reviewer's Responses to Questions

**Comments to the Author**

1. If the authors have adequately addressed your comments raised in a previous round of review and you feel that this manuscript is now acceptable for publication, you may indicate that here to bypass the “Comments to the Author” section, enter your conflict of interest statement in the “Confidential to Editor” section, and submit your "Accept" recommendation.

Reviewer #3: All comments have been addressed

2. Is the manuscript technically sound, and do the data support the conclusions?

Reviewer #3: Yes

3. Has the statistical analysis been performed appropriately and rigorously? 

Reviewer #3: Yes

4. Have the authors made all data underlying the findings in their manuscript fully available?

Reviewer #3: Yes

5. Is the manuscript presented in an intelligible fashion and written in standard English?

Reviewer #3: Yes

6. Review Comments to the Author

Reviewer #3: Overall, I believe this is an interesting study that can contribute to the literature, but needs further clarification and improvement.

1. In the introduction,

Sentence starting with “Therefore, by implementing this study..” This sentence must be come before the aim sentence.

2.Sentence starting with “In this study, we asked families …” This information can be transferred to the method section and synthesized with appropriate sentences.

3.In the method, the title of the information about ethics can be corrected as “Ethics” not “Ethnicity Criteria:"

4.When I read this article, I would like to have information about Arak, in a few short sentences.

For example, the socioeconomic status of Arak city, education, industrial city? Are the people ethnically similar? Migration level?

5.It is not clear which tests were applied in the table. a, b Which tests should be shown.

6.The limitations of the study should be given more space. As a cross-sectional study it can be difficult to establish cause-effect connections. This is an example.

7. The findings of the study after the regression analysis are actually very important findings. More insight discussion is needed on these.

Interesting findings, for example, in girls and elderly mothers have lower self-medication. Authors should provide explanations to it in their discussion.

7. PLOS authors have the option to publish the peer review history of their article (what does this mean?). If published, this will include your full peer review and any attached files.

Reviewer #3: **Yes: **Ugurcan Sayili

---

## [Author Response · Author response to Decision Letter 1]

22 Nov 2022

Reviewer #3: Overall, I believe this is an interesting study that can contribute to the literature, but needs further clarification and improvement.

In the introduction,

Sentence starting with “Therefore, by implementing this study..” This sentence must be come before the aim sentence.

Thanks for your concern, the necessary changes were implied.

Sentence starting with “In this study, we asked families …” This information can be transferred to the method section and synthesized with appropriate sentences.

Thanks for your concern, the necessary changes were implied.

In the method, the title of the information about ethics can be corrected as “Ethics” not “Ethnicity Criteria:"

Thanks for your concern, the necessary changes were implied.

When I read this article, I would like to have information about Arak, in a few short sentences.

For example, the socioeconomic status of Arak city, education, industrial city? Are the people ethnically similar? Migration level?

Arak is the capital of Markazi Province, Iran. At the 2011 census, its population was 526,182, in 160,761 families.[3][4] The city is nicknamed the "Industrial Capital of Iran". As a major industrial city, Arak hosts several industrial factories inside and within a few kilometers outside the city, including the factory of Machine Sazi Arak and the Iranian Aluminum Company. These factories produce nearly half of the needs of the country in steel, petrochemical, and locomotive industries. As an industrial city in a developing country, Arak suffers from air pollution.

Moreover, websites containing additional information useful about Arak are enlisted as follows in the references:

(1-4)

It is not clear which tests were applied in the table. a, b Which tests should be shown.

In Table 1, the results of Chi-square and Mann-Whitney U test which were used to compare mean or median values by self-medication of antibiotic consumption are depicted. In Table 2, the results of Univariate and multiple logistic regression models in order to obtain odds ratio (OR) and 95% confidence intervals is shown.

The findings of the study after the regression analysis are actually very important findings. More insight discussion is needed on these.

Interesting findings, for example, in girls and elderly mothers have lower self-medication. Authors should provide explanations to it in their discussion.

Additional explanations are added in the discussion section, thanks.

References

1. https://en.wikipedia.org/wiki/Arak,_Iran.

2. https://www.wikidata.org/wiki/Q212628.

3. https://www.macrotrends.net/cities/21484/arak/population.

4. https://www.iranicaonline.org/articles/arak.

---

## [Editor Report · Decision Letter 2]

25 Nov 2022

Prevalence and determinants of self-medication consumption of antibiotics in children in Iran: a population-based cross-sectional study, 2018-19

PONE-D-21-31099R2

Dear Dr. Moradzadeh,

We’re pleased to inform you that your manuscript has been judged scientifically suitable for publication and will be formally accepted for publication once it meets all outstanding technical requirements.

Kind regards,

Ugurcan Sayili, M.D.

Academic Editor

PLOS ONE

Additional Editor Comments (optional):

The authors have made the necessary corrections.
---

## [Editor Report · Acceptance letter]

19 Dec 2022

PONE-D-21-31099R2 

Prevalence and determinants of self-medication consumption of antibiotics in children in Iran: a population-based cross-sectional study, 2018-19 

Dear Dr. Moradzadeh:

I'm pleased to inform you that your manuscript has been deemed suitable for publication in PLOS ONE. Congratulations! Your manuscript is now with our production department. 

Kind regards, 

on behalf of

Dr. Ugurcan Sayili 

Academic Editor

PLOS ONE